# Translation, reliability, and validity of Japanese version of the Respiratory Distress Observation Scale

Hideaki Sakuramoto[1]* , Chie Hatozaki[2] , Takeshi Unoki[3‡], Gen Aikawa[2‡], Shunsuke Kobayashi[4‡], Saiko Okamoto[1‡], Shinichi Shimomura[5‡], Ayako Kawasaki[6‡], Miwako Fukui[7‡]

1 Department of Adult Health Nursing, College of Nursing, Ibaraki Christian University, Omika, Hitachi, Ibaraki, Japan, 2 Intensive Care Unit, University of Tsukuba Hospital, Amakubo, Tsukuba, Ibaraki, Japan, 3 Department of Acute and Critical Care Nursing, School of Nursing, Sapporo City University, Sapporo, Hokkaido, Japan, 4 Paediatric Intensive Care Unit, University of Tsukuba Hospital, Amakubo, Tsukuba, Ibaraki, Japan, 5 Intensive Care Unit, Tsukuba Memorial Hospital, Kaname, Tsukuba, Ibaraki, Japan, 6 Department of Nursing, Hitachi General Hospital, Jonan-cho, Hitachi, Ibaraki, Japan, 7 Intensive Care Unit, Tsukuba Medical Centre Hospital, Amakubo, Tsukuba, Ibaraki, Japan

☯ These authors contributed equally to this work.
‡ These authors also contributed equally to this work.
* gongehead@yahoo.co.jp

**Data Availability Statement:** All relevant data are within the manuscript and its Supporting Information files.

## Abstract

Dyspnea is a common, distressing symptom of cardiopulmonary and neuromuscular diseases and is defined as "a subjective experience of breathing discomfort that consists of qualitatively distinct sensations that vary in intensity." However, Japanese intensive care units (ICUs) do not routinely screen for dyspnea, as no validated Japanese version of the Respiratory Distress Observation Scale (RDOS) is available. Therefore, we aimed to translate the English version of this questionnaire into Japanese and assess its validity and reliability. To translate the RDOS, we conducted a prospective observational study in a 12-bed ICU of a universal hospital that included 42 healthcare professionals, 10 expert panels, and 128 ventilated patients. The English version was translated into Japanese, and several cross-sectional web-based questionnaires were administered to the healthcare professionals. After completing the translation process, a validity and reliability evaluation was performed in the ventilated patients. Inter-rater reliability was evaluated using Cohen's weighted kappa coefficient. Criterion validity was ascertained based on the correlation between RDOS and the dyspnea visual analog scale. The area under the receiver operating characteristic curve analysis was used to evaluate the ability of the RDOS to identify patients with self-reported dyspnea. The average content validity index at the scale level was 0.95. Data from the 128 patients were collected and analyzed. Cohen's weighted kappa coefficient and the correlation coefficient between the two scales were 0.76 and 0.443 (95% confidence intervals 0.70–0.82 and 0.23–0.62), respectively. For predicting self-reported dyspnea, the area under the receiver operating characteristic curve was 0.81 (95% confidence interval 0.67–0.97). The optimal cutoff used was 1, with a sensitivity and specificity of 0.89 and 0.61, respectively. Our findings indicated that the Japanese version of the RDOS

**Funding:** This work was supported by the Japan Society for the Promotion of Science Kakenhi (Grants-in-Aid for Scientific Research) [Grant Number 19K10859].The funders had no role in study design, data collection and analysis, decision to publish, or preparation of the manuscript.

**Competing interests:** The authors have declared that no competing interests exist.

is acceptable for face validity, understandability, criterion validity, and inter-rater reliability in lightly sedated mechanically ventilated patients, indicating its clinical utility.

## Introduction

Dyspnea is a common, distressing symptom of cardiopulmonary and neuromuscular diseases and is defined as "a subjective experience of breathing discomfort that consists of qualitatively distinct sensations that vary in intensity" [1]. In particular, intensive care unit (ICU) patients are at risk for dyspnea at different stages of their ICU stay [2–4]. In a study by Rotondi et al. [5], 22% of ICU patients recalled "not being able to get enough air through the intubation tube," and of these, 92% remembered the experience as moderately to extremely uncomfortable. Dyspnea occurs in 47% of patients undergoing mechanical ventilation [4, 6] and is associated with adverse patient outcomes, such as prolonged ventilation, anxiety, noninvasive ventilation failure, and mortality [2–4].

Despite the essential cooperation of the patients in assessing the presence and intensity of dyspnea [1], most ICU patients are unable to self-report due to lack of consciousness owing to, for example, cognitive impairment, delirium, and use of sedatives. Indeed, it has been reported that approximately 80% of patients experience delirium during their ICU stay [7]. Until recently, all dyspnea assessment tools used in Japan have been designed for conscious patients who can express themselves. In other countries, the Respiratory Distress Observation Scale (RDOS) developed in 2008 has shown good reliability and validity in patients who are unable to self-report dyspnea, such as those in palliative care or with chronic obstructive pulmonary disease, heart failure, and pneumonia [8]. Additionally, recent studies have validated the RDOS as a surrogate measure of self-reported dyspnea in critically ill patients [9]. Therefore, the RDOS is considered to be useful for the assessment of dyspnea in the ICU setting.

However, Japanese ICUs do not routinely screen for dyspnea, as no validated Japanese version of the RDOS is available for use in critically ill patients. Without validated tools, dyspnea may be neglected and go untreated in Japanese ICU patients. Therefore, this study aimed to translate the RDOS into Japanese and evaluate its validity and reliability.

## Materials and methods

### Design

The original RDOS was translated into Japanese, and several cross-sectional web-based questionnaires were administered to healthcare professionals to assess their content validity. Thereafter, validation and reliability studies were conducted to evaluate the criterion-related validity and inter-rater reliability.

### Translation process

After receiving written permission from the original author, Dr. Campbell [8], the RDOS translation was commenced using the back-translation method based on a translation, adaptation, and validation guideline for scales [10] to ensure linguistic and cultural equivalence. The translation team was composed of three nurse educators and four clinical nurses. The translation process was conducted in four steps, also including an assessment of content validity.

Step 1: Forward translation of the RDOS from English to Japanese was performed independently by a nurse educator and an expert nurse. The two translations were synthesized into

one preliminary version through discussion between both translators to assess the equivalence of meaning between the original and translated versions.

Step 2: Two translators who were blinded to the original RDOS translated the forward-translated version of the RDOS from Japanese into English. One translator was knowledgeable about health terminology and the content area of the construct of the instrument in English. The other translator was knowledgeable about the cultural and linguistic nuances of English. The two back-translated versions of the RDOS and the original RDOS were compared by multidisciplinary experts to assess the accuracy of terminology and clarity of expressions in terms of consistency in meaning.

Step 3: To evaluate the clarity of the items, a pilot test of the Japanese version of the RDOS was conducted. A total of 42 healthcare professionals (27 nurses, 10 doctors, and 3 certified critical care nurses) evaluated each item of the RDOS using a dichotomous scale ("easy to understand" vs. "difficult to understand") via a web-based questionnaire. The percentage of "unknown" for each item was calculated, and items that were evaluated as "unknown" by >20% of the evaluators were revised.

Step 4: The expert panel consisted of eight nurses, one physical therapist, and two physicians with expertise in respiratory care, who evaluated the clarity of the RDOS using the same process as in Step 3. Thereafter, each item was evaluated for content validity by an expert panel on a four-point Likert scale (1 = not relevant; 2 = unable to assess relevance; 3 = relevant but needs minor alterations; and 4 = very relevant and succinct). The average content validity index at the scale level (S-CVI/Ave) was calculated. An S-CVI/Ave of ≥0.90 was considered acceptable [10].

## Criterion-related validity and inter-rater reliability of the Japanese version of the RDOS

A validation and reliability study was conducted in a 12-bed medical-surgical ICU of a universal hospital between February 2020 and February 2021.

## Patients and recruitment process

We enrolled adult ICU patients who had received mechanical ventilation for ≥24 h. Patients who were expected to die within 48 h, had already received ≥24 h of mechanical ventilation prior to ICU admission, had a history of mental illness, did not understand Japanese, received neuromuscular blocking agents, or had paralysis/neuromuscular disorders were excluded. This study was approved by the Institutional Review Board of the Study Coordinator Center (Ibaraki Christian University; approval number 2019–013) and the Ethics Board of the University of Tsukuba Hospital Research Ethics Review Committee (approval number R01-184). Written informed consent was obtained from patients or their relatives according to the Institutional Review Board recommendations after providing the participants with a detailed description of the study.

## Data collection

Baseline characteristics were recorded, including age, sex, diagnosis for ICU admission, ventilation status, sedative use, opiate use, and disease severity, which was calculated using the Acute Physiology and Chronic Health Evaluation II. A pair consisting of a researcher and an untrained nurse simultaneously and independently assessed an ICU patient with regards to

the RDOS, depth of sedation levels using the Richmond Agitation–Sedation Scale, delirium using the confusion assessment tool for ICUs [11], and pain using the Behavioral Pain Observational Scale. Additional evaluation using the observational dyspnea visual analog scale (D-VAS) [9, 12] was performed objectively by a trained nurse. A correlation coefficient of <0.20 was considered as a "slight, almost negligible relationship," in the range 0.20–0.40 as a "low correlation," 0.40–0.70 as a "moderate correlation," 0.70–90 as a "high correlation," and >0.90 as a "very high correlation." The main researcher was blinded to the scores of the others, and evaluation with the D-VAS was performed before the RDOS to remove any bias.

## Sample size

The required sample size was calculated based on reliability, as previously described [9]. Inter-observer agreement based on previous studies was used as an estimate of moderate correlation (r = 0.44–0.76) [9, 13]. A sample size of 17–38 patients was determined to be sufficient at a significance level ($\alpha$) of 0.05 and a test power (1-$\beta$) of 0.90 [14].

## Data analysis

The data are expressed as numbers and percentages or as medians and interquartile ranges for nonparametrically distributed data or as means and standard deviations for parametrically distributed data. Inter-rater reliability was evaluated using a weighted Cohen's kappa coefficient and Spearman's rank-order correlations. Criterion validity was ascertained using Spearman's rank-order correlations to examine the association between the RDOS and the D-VAS. The area under the receiver operating characteristic curve analysis was used to evaluate the diagnostic ability of the RDOS in identifying patients with self-reported dyspnea. The sensitivity and specificity for various RDOS cutoff points were calculated, and the Youden index was utilized to detect the optimal RDOS score cutoff for our sample. This was then compared with the original cutoff of the RDOS of ≥3 [13]. All analyses were conducted using with SPSS Statistics 25.0.

## Results

### Translation

The back-translated version of the RDOS was submitted to the original author for approval and then evaluated by 42 healthcare professionals. The item of "Restlessness: movement without purpose" was revised because 31% of the participants evaluated the item as "unclear." A panel of ten experts critiqued the translated RDOS and three items were evaluated as "unclear" by >20% of the panel; the translation team thus revised these items. The S-CVI/Ave was 0.95, indicating acceptable content validity. Thus, the Japanese version of the RDOS was finalized (S1 Fig; the individual in this RDOS scale photograph has given written informed consent (as outlined in the PLOS consent form) to publish these case details).

### Patient characteristics

Of the 719 patients admitted to the ICU, 591 were excluded (Fig 1). The data of 128 patients were collected, and their characteristics are presented in Table 1. Each participant was evaluated either one or two times. In total, 213 observations were made. In 123/213 (65%) of these observations, the patient was able to communicate. Self-reported dyspnea was present in 23/213 patients (10.8%), and the mean D-VAS score was 0.61 (1.9). The mean RDOS score was 0.98 (1.2), and 22/213 patients (10.3%) had dyspnea with an RDOS score of >3.

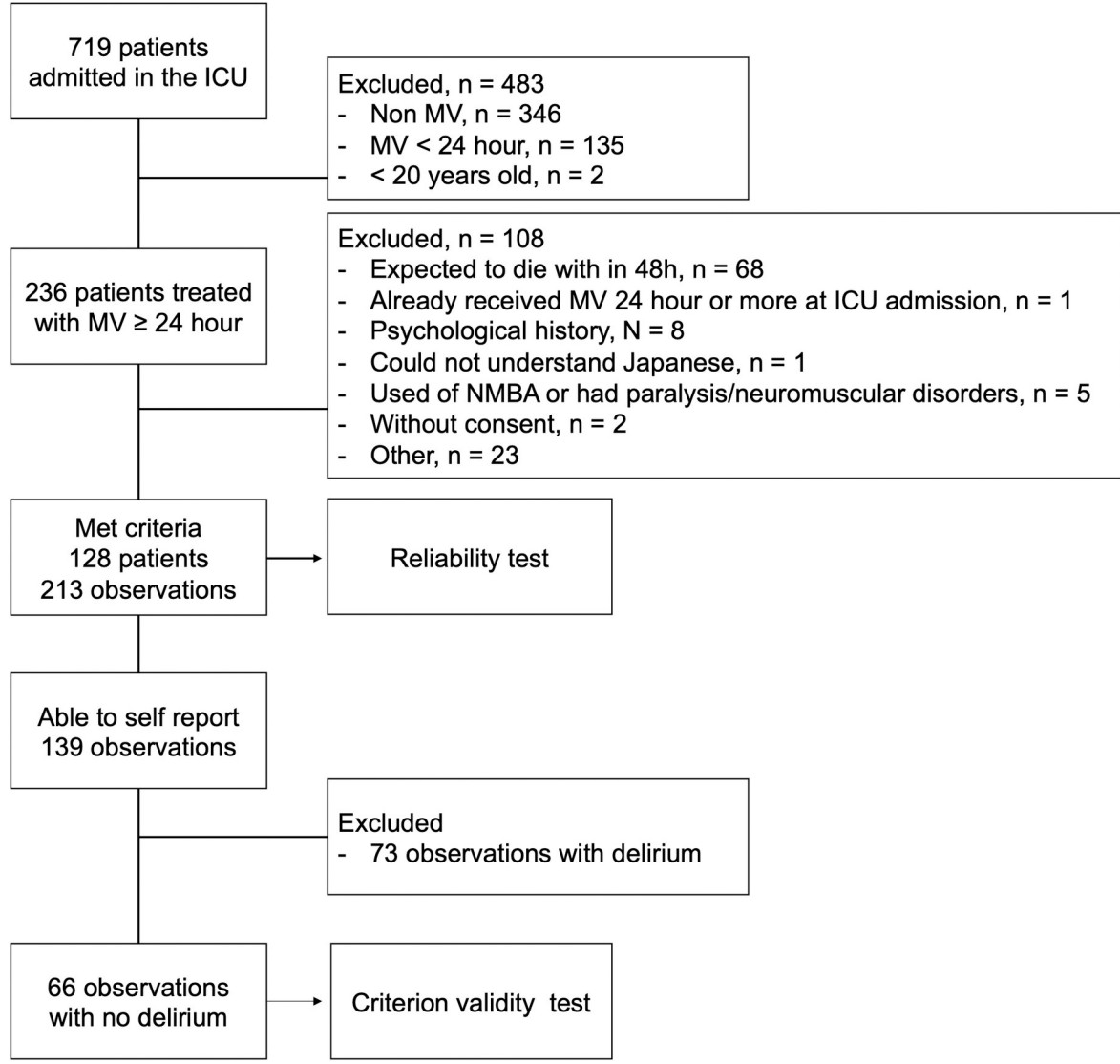

**Fig 1. Flowchart of the patient recruitment process.** MV, mechanical ventilation; NMBA, neuromuscular blocking agent.

### Inter-rater reliability

In the 213 paired observations by the researcher and nurse, both RDOS scores showed excellent inter-rater reliability with a weighted Cohen's kappa coefficient of 0.76 (95% confidence interval [CI] 0.70–0.82). The correlation of the RDOS scores measured by the researcher and nurse was 0.82 (95% CI 0.76–0.87), representing a high correlation and therefore, good inter-rater reliability.

### Criterion validity

Of the 213 paired observations, criterion validity was evaluated in 66 observations excluding those in patients with delirium and/or in patients who were unable to self-report dyspnea. The correlation coefficient between the RDOS and the D-VAS was 0.443 (95% CI 0.23–0.62). For the prediction of self-reported dyspnea, the RDOS area under the receiver operating

**Table 1. Characteristics of 213 observations in critically ill ventilated patients.**

| Characteristics | Overall *n = 213* |
|---|---|
| Age, years, mean (SD) | 66.5 (13.4) |
| Male, n (%) | 149 (69.6) |
| BMI, kg/m², mean (SD) | 25.0 (5.9) |
| APACHE II, mean (SD) | 19/1 (7.6) |
| Surgery, n (%) | 148 (69.2) |
| Unplanned ICU admission, n (%) | 142 (66.4) |
| Cause of ICU admission | |
| Heart failure, n (%) | 105 (49.1) |
| Respiratory failure, n (%) | 16 (7.5) |
| Gastrointestinal disease, n (%) | 36 (16.8) |
| Sepsis, n (%) | 9 (4.2) |
| Other, n (%) | 48 (22.4) |
| Duration of mechanical ventilation, days, mean (SD) | 6.8 (8.8) |
| Length of ICU stay, days, mean (SD) | 10.4 (9.3) |
| Length of hospital stay, days, mean (SD) | 36.1 (25.5) |
| Pain and sedation status | |
| Administration of analgesia, yes, n (%) | 165 (77.1) |
| Administration of sedatives, no, n (%) | 67 (31.5) |
| RASS, median (IQR) | −1 (−4, −1) |
| BPS, median (IQR) | 3 (3, 3) |

APACHE II, Acute Physiology and Chronic Health Evaluation II; BMI, body mass index; BPS, Behavioral Pain Observational Scale; ICU, intensive care unit; IQR, interquartile range; RASS, Richmond Agitation-Sedation Scale; SD, standard deviation.

characteristic curve was 0.81 (95% CI 0.67–0.97) (Fig 2). The optimal cutoff in this study was 1 with a sensitivity and specificity of 0.89 and 0.61, respectively. The sensitivity and specificity of the original cutoff score of 3 were 0.56 and 0.95, respectively.

## Discussion

This study is the first to formally translate the RDOS from English to Japanese using a multi-step back-translation method based on a guideline specific to the translation, adaptation, and validation of scales [10]. The Japanese version of the RDOS was found to be acceptable in terms of face validity, understandability, criterion validity, and inter-rater reliability in lightly sedated mechanically ventilated patients. These results indicate that the Japanese version of the RDOS is acceptable for clinical use.

Translation was performed according to the standard method, and face validity, relevant validity, and understandability were established. As pointed out in a previous study, direct translation does not guarantee sufficient equivalence [10]; therefore, we used the back-translation method to correct for content variation. Back-translation by two translators with different backgrounds was performed in consideration of the differences in medical terminology and subtle nuances. Finally, face validity, relevant validity, and understandability were evaluated and established by multidisciplinary experts.

We confirmed high inter-rater reliability. These findings are consistent with those of previous studies [8, 9] and indicate that the RDOS was used accurately by every nurse. We consider that not only the text but also the pictures on how to assess RDOS likely contributed to this

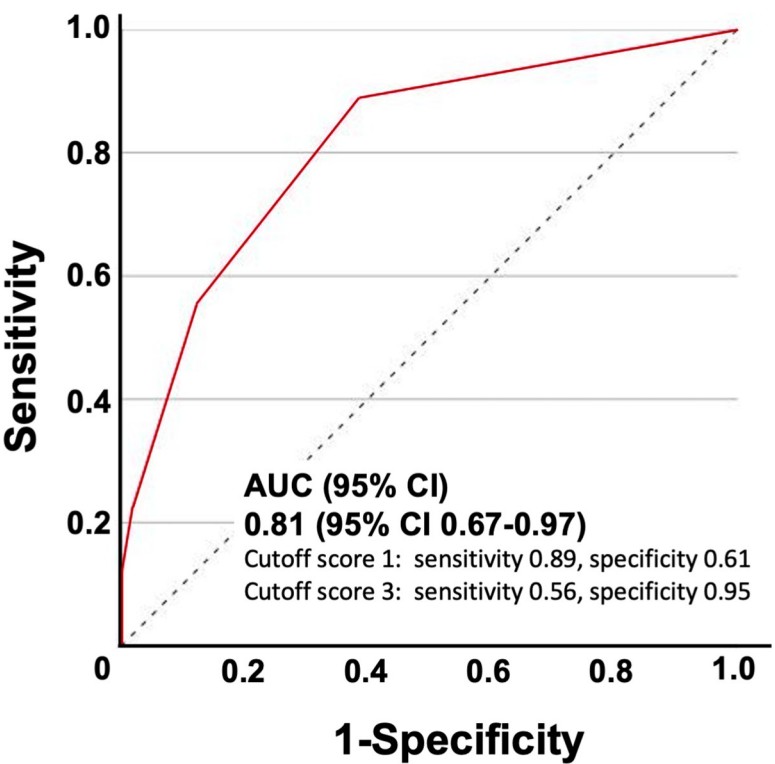

**Fig 2. Prediction of self-reported dyspnea in patients without delirium using the Respiratory Distress Observation Scale.** AUCs for the Respiratory Distress Observation Scale as a predictor of self-reported dyspnea in patients without delirium. For the prediction of self-reported dyspnea, the RDOS AUC was 0.81 (95% CI 0.67–0.97). The optimal cutoff was 1 in this study, with sensitivity = 0.89 and specificity = 0.61. For the original cutoff score of 3, the sensitivity was 0.56 and specificity was 0.95. AUC, Areas under the receiver operating characteristic curves; 95% CI, 95% confidence interval.

accuracy. Furthermore, in this study, one evaluator was not trained for using the RDOS; however, high inter-rater reliability was obtained. Accordingly, the Japanese version of the RDOS is easy to use regardless of training and can be easily performed within 1–3 min.

The criterion validity was found to be within an acceptable range. These findings are consistent with those of previous studies [9, 13]. Thus, the Japanese version of the RDOS was translated accurately and validated. Additionally, the Japanese version of the RDOS may indicate the severity of dyspnea. Thus, clinicians may be able to quantitatively evaluate dyspnea, which is important for observing the response to changes in mechanical ventilation settings and palliative care.

The cutoff point of the Japanese version of the RDOS was 1, while the cutoff point of the original version was 3. There are two potential reasons for this difference. First, translation issues may not be valid. However, this is unlikely considering that the translation process was in accordance with guidelines and the criterion validity was acceptable. Second, differences in language, ethnicity, and culture can alter the perception and expression of the sensation of dyspnea [15]. Japanese people, in particular, exhibit reduced facial emotional expression [16]. In addition, in this study, about 70% of the participants were male. Considering that the percentage of males in past surveys was also 70%, it is unlikely that there was any influence of sex on the cutoff or other factors. Further, this study included ventilated patients, therefore the effects of sedation may have caused less expression in the patient's face.However, it was not our objective to determine the cutoff point, and the sample size was too small for this purpose. To confirm the optimal cutoff point, further studies with larger sample sizes are needed.

### Clinical and research implications

The validated Japanese version of the RDOS is available for use in critically ill patients and thus should be included in their routine care [17]. The RDOS is feasible because it is easy to use and requires no training. Moreover, we believe that the scientific procedures and processes used for language translation as presented in our study can be used to conduct similar research for other languages in future studies.

### Limitations

This study had several limitations. First, the Japanese version of the RDOS was evaluated in patients who were able to self-report. Ideally, the validity of the RDOS should be evaluated in patients who cannot self-report. However, this seems to be methodologically difficult. Second, the participants were ventilated patients in a single medical and surgical ICU, leading to an inherent risk of data overfitting. Moreover, the validity was not evaluated in nonventilated patients. Therefore, the results may not be generalizable to all ICU patients. However, the original RDOS has been validated in nonventilated patients; thus, the Japanese version is likely to also be useful in such patients.

## Conclusion

The Japanese version of the RDOS is acceptable for face validity, understandability, criterion validity, and inter-rater reliability in lightly sedated mechanically ventilated patients, indicating its clinical utility. Further studies with larger sample sizes are needed to confirm the cutoff point of the Japanese version of the RDOS.

## Supporting information

**S1 Fig. Japanese version of the Respiratory Distress Observation Scale.**
(TIF)

**S1 Dataset. Minimal dataset.**
(XLSX)

## Acknowledgments

We are grateful to Dr. Margaret L. Campbell for granting us permission to translate the RDOS and providing guidance. We would also like to thank Dr. Ryuichi Hasegawa, Dr. Yuki Enomoto, Mr. Tsuyoshi Maruyama, Ms. Megumi Moriyasu, Ms. Masako Shirasaka, Ms. Yasuyo Yoshino, Ms. Kumi Sunaoshi, Mr. Takahiro Tsujimoto, Mr. Tamoto Mitsuhiro, Ms. Mio Kitayama, and Mr. Kohsuke Sakaki for their expert comments. We would like to thank Editage (www.editage.com) for English language editing.

## Author Contributions

**Conceptualization:** Hideaki Sakuramoto.

**Data curation:** Hideaki Sakuramoto, Chie Hatozaki.

**Formal analysis:** Hideaki Sakuramoto, Chie Hatozaki, Takeshi Unoki.

**Funding acquisition:** Hideaki Sakuramoto.

**Investigation:** Hideaki Sakuramoto, Chie Hatozaki, Gen Aikawa, Shunsuke Kobayashi, Shinichi Shimomura, Ayako Kawasaki, Miwako Fukui.

**Methodology:** Hideaki Sakuramoto, Chie Hatozaki, Takeshi Unoki.

**Project administration:** Hideaki Sakuramoto, Takeshi Unoki, Saiko Okamoto, Shinichi Shimomura, Ayako Kawasaki, Miwako Fukui.

**Validation:** Hideaki Sakuramoto, Chie Hatozaki.

**Writing – original draft:** Hideaki Sakuramoto.

**Writing – review & editing:** Chie Hatozaki, Takeshi Unoki, Gen Aikawa, Shunsuke Kobayashi, Saiko Okamoto, Shinichi Shimomura, Ayako Kawasaki, Miwako Fukui.

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
