## [Decision Letter · Decision Letter 0]

9 Jul 2021

PONE-D-21-17304

Translation, reliability, and validity of the Japanese version of the Respiratory Distress Observation Scale

PLOS ONE

Dear Dr. Sakuramoto,

Thank you for submitting your manuscript to PLOS ONE. After careful consideration, we feel that it has merit but does not fully meet PLOS ONE’s publication criteria as it currently stands. Therefore, we invite you to submit a revised version of the manuscript that addresses the points raised during the review process.

We look forward to receiving your revised manuscript.

Kind regards,

Tai-Heng Chen, M.D.

Academic Editor

PLOS ONE

2. We note that Figure S1 includes an image of a participant in the study. 

As per the PLOS ONE policy (http://journals.plos.org/plosone/s/submission-guidelines#loc-human-subjects-research) on papers that include identifying, or potentially identifying, information, the individual(s) or parent(s)/guardian(s) must be informed of the terms of the PLOS open-access (CC-BY) license and provide specific permission for publication of these details under the terms of this license.

Please download the Consent Form for Publication in a PLOS Journal (http://journals.plos.org/plosone/s/file?id=8ce6/plos-consent-form-english.pdf). The signed consent form should not be submitted with the manuscript, but should be securely filed in the individual's case notes.

Please amend the methods section and ethics statement of the manuscript to explicitly state that the patient/participant has provided consent for publication: “The individual in this manuscript has given written informed consent (as outlined in PLOS consent form) to publish these case details”.

Reviewers' comments:

Reviewer's Responses to Questions

**Comments to the Author**

1. Is the manuscript technically sound, and do the data support the conclusions?

Reviewer #1: Yes

Reviewer #2: Yes

2. Has the statistical analysis been performed appropriately and rigorously? 

Reviewer #1: Yes

Reviewer #2: Yes

3. Have the authors made all data underlying the findings in their manuscript fully available?

Reviewer #1: Yes

Reviewer #2: Yes

4. Is the manuscript presented in an intelligible fashion and written in standard English?

Reviewer #1: Yes

Reviewer #2: Yes

5. Review Comments to the Author

Reviewer #1: The paper “Translation, reliability, and validity of the Japanese version of the Respiratory Distress Observation Scale” by Hideaki Sakuramoto et al. presents the process of translation and validation of an important dyspnoea scale in Japanese language. The authors follow the standard rules related to this topic with a clear methodology, cohorts, results and discussion.

There are some minor issues that can be solved.

1. Page 4, translation process, line 91. “The RDOS translation was commenced using the back-translation method based on a translation, adaptation, and validation guideline for scales…”

Do you have a translation by a professional certified company?

2. Page 8, line 191. Table 1. Approx. 70% of patients are female. Is there a limitation? Can you comment at limitation?

3. Figure 2. The sensitivity, specificity, positive predictive value, and negative predictive value for various RDOS cut-off points. It needs short explanation for results on the figure.

4. Is there a possibility to make it available online for Japanese speaking medical personnel?

Reviewer #2: Japanese ICUs do not routinely screen for dyspnea, as no validated

Japanese version of the RDOS is available for use in critically ill patients. Therefore, this study aimed to translate the RDOS into Japanese and evaluate its validity and reliability.

This is an interesting study, well conducted and written.

I suggest the authors to include the area under the receiver operating

characteristic curve, with 95% CI, in Figure 2.

6. PLOS authors have the option to publish the peer review history of their article (what does this mean?). If published, this will include your full peer review and any attached files.

Reviewer #1: **Yes: **Mihaicuta Stefan

Reviewer #2: No

---

## [Author Response · Author response to Decision Letter 0]

18 Jul 2021

Response to Reviewers

Thank you for pointing this out. We have ensured that our manuscript follows PLoS One’s guidelines. 

2. We note that Figure S1 includes an image of a participant in the study. 

As per the PLOS ONE policy (http://journals.plos.org/plosone/s/submission-guidelines#loc-human-subjects-research) on papers that include identifying, or potentially identifying, information, the individual(s) or parent(s)/guardian(s) must be informed of the terms of the PLOS open-access (CC-BY) license and provide specific permission for publication of these details under the terms of this license.

Please download the Consent Form for Publication in a PLOS Journal (http://journals.plos.org/plosone/s/file?id=8ce6/plos-consent-form-english.pdf). The signed consent form should not be submitted with the manuscript, but should be securely filed in the individual's case notes.

Please amend the methods section and ethics statement of the manuscript to explicitly state that the patient/participant has provided consent for publication: “The individual in this manuscript has given written informed consent (as outlined in PLOS consent form) to publish these case details”.

Thank you for pointing this out. Consent has been obtained from the subject in the photo. We obtained a signature in the Consent Form for Publication provided by the PLOS Journal as you have pointed out; we have safely retained it in the individual’s case notes.

Since there was no appropriate place to include the above information in the Methods section of the manuscript, we have clearly indicated it next to the supplementary figure, S1 Fig’s citation in the Results section; the individual in the RDOS scale photograph has consented to publication according to the guidelines mentioned in the PLOS ONE form. Please see Page 7, Line 180 for the corresponding revision.

" The individual in the RDOS scale photograph has given written informed consent (as outlined in PLOS consent form) to publish these case details."

Comments to the Author

Reviewer #1

1. Page 4, translation process, line 91. “The RDOS translation was commenced using the back-translation method based on a translation, adaptation, and validation guideline for scales…” Do you have a translation by a professional certified company?

We do not have a translation by a professional accreditation company; however, we have followed the several procedures specified in the guidelines for translating scientific tools between different cultures and languages. We have followed the steps in the cited reference, as mentioned below, for translation.

Sousa VD, Rojjanasrirat W. Translation, adaptation and validation of instruments or scales for use in cross-cultural health care research: a clear and user-friendly guideline. J Eval Clin Pract. 2011;17(2):268-74. Epub 2010/09/30. doi: 10.1111/j.1365-2753.2010.01434.x. PubMed PMID: 20874835.

2. Page 8, line 191. Table 1. Approx. 70% of patients are female. Is there a limitation? Can you comment at limitation?

Thank you for the important remarks. We have checked the data again and found that it was a simple notation mistake; we had about 70% males. The ratio of males to females in the previous study was similar (with about 70% males), and we did not think that it would have a significant impact on the results. We have also added this point to our Discussion (please see Table 1 and Pages 10–11, Lines 255–258).

3. Figure 2. The sensitivity, specificity, positive predictive value, and negative predictive value for various RDOS cut-off points. It needs short explanation for results on the figure.

A short explanation, as well as the results of sensitivity and sensitivity at different cutoff points and the area under the receiver operating characteristic curve with 95% CI, have been added to the figure legend (please see Figure 2 and Page 9, Lines 219–222).

4. Is there a possibility to make it available online for Japanese speaking medical personnel?

Thank you very much for your idea. If it gets published, we will make it available online as a supplemental file (S1 Fig) in PLOS ONE and on our lab's website (please see S1 Fig file).

Reviewer #2: 

1. I suggest the authors to include the area under the receiver operating characteristic curve, with 95% CI, in Figure 2.

Can you discuss any measures to further reduce rates of PTSD and improve quality of life post ICU discharge?

We have followed your suggestion and added the area under the receiver operating characteristic curve, with 95% CI, to Figure 2.

---

## [Decision Letter · Decision Letter 1]

28 Jul 2021

Translation, reliability, and validity of the Japanese version of the Respiratory Distress Observation Scale

PONE-D-21-17304R1

Dear Dr. Sakuramoto,

We’re pleased to inform you that your manuscript has been judged scientifically suitable for publication and will be formally accepted for publication once it meets all outstanding technical requirements.

Kind regards,

Tai-Heng Chen, M.D.

Academic Editor

PLOS ONE

Reviewers' comments:

Reviewer's Responses to Questions

**Comments to the Author**

1. If the authors have adequately addressed your comments raised in a previous round of review and you feel that this manuscript is now acceptable for publication, you may indicate that here to bypass the “Comments to the Author” section, enter your conflict of interest statement in the “Confidential to Editor” section, and submit your "Accept" recommendation.

Reviewer #1: All comments have been addressed

Reviewer #2: All comments have been addressed

2. Is the manuscript technically sound, and do the data support the conclusions?

Reviewer #1: Yes

Reviewer #2: Yes

3. Has the statistical analysis been performed appropriately and rigorously? 

Reviewer #1: Yes

Reviewer #2: Yes

4. Have the authors made all data underlying the findings in their manuscript fully available?

Reviewer #1: Yes

Reviewer #2: Yes

5. Is the manuscript presented in an intelligible fashion and written in standard English?

Reviewer #1: Yes

Reviewer #2: Yes

6. Review Comments to the Author

Reviewer #1: The authors responded to all issues. There are some questions about the ability of the authors to perform a 100% right translation without the assistance of a professional certified company, but the process of translating the document fulfilled all professional criteria. In my opinion this is good enough for a medical document.

Reviewer #2: The authors answered all questions, and included the AUC in Figure.

All comments have been addressed.

7. PLOS authors have the option to publish the peer review history of their article (what does this mean?). If published, this will include your full peer review and any attached files.

Reviewer #1: **Yes: **Stefan Mihaicuta

Reviewer #2: No

---

## [Editor Report · Acceptance letter]

2 Aug 2021

PONE-D-21-17304R1 

Translation, reliability, and validity of Japanese version of the Respiratory Distress Observation Scale 

Dear Dr. Sakuramoto:

I'm pleased to inform you that your manuscript has been deemed suitable for publication in PLOS ONE. Congratulations! Your manuscript is now with our production department. 

Kind regards, 

on behalf of

Dr. Tai-Heng Chen 

Academic Editor

PLOS ONE